# ECSS: High-Embedding-Capacity Audio Watermarking with Diversity Reception

**DOI:** 10.3390/e24121843

**Published:** 2022-12-17

**Authors:** Shiqiang Wu, Ying Huang, Hu Guan, Shuwu Zhang, Jie Liu

**Affiliations:** 1School of Artificial Intelligence, University of Chinese Academy of Sciences, Beijing 101408, China; 2Institute of Automation, Chinese Academy of Sciences, Beijing 100045, China

**Keywords:** digital audio watermarking, embedding capacity, spread spectrum, diversity reception

## Abstract

Digital audio watermarking is a promising technology for copyright protection, yet its low embedding capacity remains a challenge for widespread applications. In this paper, the spread-spectrum watermarking algorithm is viewed as a communication channel, and the embedding capacity is analyzed and modeled with information theory. Following this embedding capacity model, we propose the extended-codebook spread-spectrum (ECSS) watermarking algorithm to heighten the embedding capacity. In addition, the diversity reception (DR) mechanism is adopted to optimize the proposed algorithm to obtain both high embedding capacity and strong robustness while the imperceptibility is guaranteed. We experimentally verify the effectiveness of the ECSS algorithm and the DR mechanism, evaluate the performance of the proposed algorithm against common signal processing attacks, and compare the performance with existing high-capacity algorithms. The experiments demonstrate that the proposed algorithm achieves a high embedding capacity with applicable imperceptibility and robustness.

## 1. Introduction

With the booming development of the mobile Internet, information dissemination has changed. Internet firms such as Netflix, YouTube, and Apple have replaced those outdated CD stores—fewer and fewer music is traded via CDs and tapes. This much easier dissemination has led to a prosperous music market. However, beneath the appearance of prosperity, copyright concerns have been getting worse—a piece of music can be copied and distributed too quickly on the Internet. According to the IFPI (International Federation of the Phonographic Industry) report [1], almost a third (29%) of respondents admitted using unlicensed or illegal sources to listen to or download music. There are laws in place to protect music copyrights, but determining infringement remains a challenge.

Engineers and researchers have also come up with several technologies, among which digital audio watermarking is quite representative [2], as a technology that can be used to protect the copyright of music or conference recordings. Digital audio watermarking embeds the copyright identifier, also known as a watermark, imperceptibly into the host audio. When a copyright conflict occurs, it can be verified by extracting the watermark from the watermarked audio.

The performance of a watermarking algorithm is measured by three main criteria: imperceptibility, robustness and embedding capacity. The **imperceptibility** refers to the data fidelity of the watermarked audio, indicating that the embedded watermarks can only introduce perceptually indistinguishable distortion to the host audio; the **robustness** refers to the reliability of the watermark that can be effectively extracted when watermarked audio has undergone some attacks, evaluated by the accuracy of the watermark extraction; the **embedding capacity** refers to the amount of watermark information that can be extracted from the watermarked audio [3,4]. The embedding capacity actually describes the ability of watermarking algorithm to transmit watermark information, which is associated with the two stages of embedding and extraction. Even some attacks that damage the robustness will reduce the capacity of the algorithm.

As copyright content becomes diverse, copyright owners want to embed the licensed object, the class of license and the start and end date into the host data, leading to the embedding capacity or payload gradually becoming a bottleneck for the watermarking system. Several studies have been conducted on the embedding capacity of audio watermarking algorithms, and they use the number of watermark bits embedded per second (bps) as a metric for the embedding capacity. Nevertheless, it relates to the sample rate or the channels of the host audio and does not fully reflect the performance of watermarking algorithms. Moreover, the imperceptibility and robustness have substantial effects on the embedding capacity due to the **embedding strength**, a crucial parameter in watermarking algorithms, controlling different performance criteria. Intuitively, the relaxed imperceptibility and strong robustness can lead to a high embedding capacity. However, existing studies lack the theoretical analysis of the embedding capacity and its dependency on other performance criteria. Therefore, improving the embedding capacity of watermarking algorithms still remains an open problem.

In communication systems and watermarking systems, there are numerous paired concepts, such as the transmitter versus the embedding stage and the receiver versus the extraction stage. Actually, communication systems are a significant influence on many watermarking systems [5,6,7,8]. The embedding capacity and channel capacity in communications are also such paired concepts, which both measure the amount of information flowing through their systems. Indeed, we examine the embedding capacity from the communication perspective in this paper.

The spread-spectrum (SS)-based watermarking system [5,9,10], the most representative watermarking algorithms, are directly derived from communications. Analyzing the embedding capacity of the SS-based watermarking algorithm from an information-theoretic perspective, we discover that increasing the embedding codebook size and the watermark extraction precision results in a higher embedding capacity. Therefore, we propose an extended-codebook spread-spectrum (ECSS) watermarking algorithm using diversity reception (DR) [11]. The contributions of this paper can be summarized as follows.

We construct the capacity model of the SS-based watermarking algorithm, which reveals how the embedding capacity is dependent on codebook size and robustness.We propose the ECSS algorithm and analyze its embedding capacity and the accuracy of the watermark extraction. By extending the codebook, each sequence can carry more than one watermark bit, which improves the embedding capacity with applicable watermark robustness.We adopt a diversity reception mechanism for optimizing the ECSS. This mechanism repeatedly embeds the watermark into different regions of the host audio and aggregates the HSI from all repeated regions when extracting the watermark, reducing the randomness of the HSI and achieving a higher embedding capacity and stronger robustness.

The remainder of this paper is organized as follows. Section 2 describes the works related to embedding capacity on watermarking and information hiding. Section 3 introduces the communication model of SS-based algorithms and formulates the embedding capacity. Section 4 proposes the ECSS algorithm and analyzes the embedding capacity and the accuracy of the watermark extraction. Section 5 introduces the DR mechanism and Section 6 presents the specific watermarking system (In this paper, two terms, watermarking algorithm and system, are used. The former is concerned with the mathematical model of the embedding and extraction stages, while the latter is the concrete implementation, which also contains modules such as the embedding location selection and the domain transformation.) flow. Section 7 is devoted to experiments and related discussions. The work of this paper is summarized in Section 8.

## 2. Related Works

Researchers have proposed numerous algorithms to heighten the embedding capacity of audio watermarking, which focus on proposing novel multibit embedding approaches. These algorithms can be roughly categorized into QIM-based (quantization index modulation), and SS-based embedding approaches. QIM-based embedding approaches achieve the effectiveness of multibit embedding by delicately constructing the quantized features or using more quantization levels. SVD-A-QIM [12] converted audio clips into feature matrices and embedded the watermark into the singular values of matrices; Wang [13] observed that the phase of an audio frame distributed uniformly in [0,2π] and applied the phase shifting for watermarking. SS-based watermarking algorithms construct an ensemble of pseudorandom sequences to express multibit watermarks or choose shorter sequences to heighten the embedding capacity. Xiang adopted circularly shifted pseudonoise codes [14] or orthogonal pseudorandom sequences [15] to implement multibit embedding approaches. Zhang [16] used a correlation-and-bit-aware embedding approach to shorten the vector length required for each bit. All these efforts to improve the embedding capacity have been successful from the technical level, but there is a lack of theoretical analysis of the concept of embedding capacity. Since the SS-based algorithms are more similar to the communication system in detail, this paper employs information theory to analyze the SS algorithm.

Information theory, which provides several theoretical foundations, is widely used in the design of watermarking algorithms [6,17,18,19]. These watermarking algorithms model the HSI and attacks as a communication channel whose output can be characterized by a conditional probability. The distribution of the host data in the transform domain is often assumed to be a generalized Gaussian distribution [20] or a Weibull distribution [21]. One of the most influential works on embedding capacity [6] treated the embedding stage as encoding processing, while the attacks that audio undergoes were treated as a communication channel $A(\hat{y}\left|y\right)$. If the attacks could be explicitly described as a conditional probability distribution, this would provide theoretical guidance for designing a suitable encoding (or embedding) processing. However, researchers often know little about the attacks, or the attacks are too complex to be modeled statistically. In addition, information theory is also broadly applied in the design of information hiding, and some entropy encodings are often used to improve the performance of information-hiding algorithms [22,23,24]. Meanwhile, communication channels have also been applied to model embedding and extraction approaches to analyze the information-hiding capacity. Fridrich [25] presented a capacity model for information hiding for a given embedding distortion and transmission distortion, which established three channels for embedding, extraction and transmission and derived the steganographic capacity as the difference of the mutual information between the codebook and the cover data. Due to the difficulty of calculating mutual information for data with a complex distribution, Fridrich’s model finally provided only an approximate result, which made it hard for inspiring the design of an information-hiding algorithm. Moreover, several works investigated information-hiding algorithms and their capacity, and these works commonly used Gaussian distribution [26,27], quantized Gaussian distribution [28], generalized Gaussian model [29], Gaussian Markov field [30] or Gibbs distribution [31] to model the host data. These works focused on the probability of steganographic messages being discovered by the warden and ignored the effect of the accuracy of the extraction, thus making the conclusions of their analysis challenging to generalize to audio watermarking systems.

In this paper, the embedding capacity of the SS-based watermarking algorithm is analyzed using information theory. Based on the analysis, we propose an extended-codebook spread-spectrum watermarking (ECSS) algorithm which enhances the embedding capacity, and finally, we adopt a diversity reception mechanism to further optimize the embedding capacity and robustness of ECSS.

## 3. Embedding Capacity of SS-Based Algorithms

As the first work to apply spread spectrum to digital watermarking, vanilla spread spectrum (VSS) has had a significant influence thanks to the anti-interference capability of the spread spectrum [32]. It is assuming x,y and p are three equal-length row vectors. Among of them, x is the **signal vector** of the host audio in a particular domain, y is its corresponding signal vector of watermarked audio, and p is a **pseudorandom sequence** independent of x. Throughout this paper, the length of signal vector x is used to denote the length of these three vectors. The VSS embeds a sign ω∈{−1,+1} into the host signal x as
(1)y=x+αωp
where α, named embedding strength, is a small positive constant controlling the embedding distortion. Notice that the watermark bits usually consists of 0’s and 1’s. Typically, the 0’s are replaced with −1’s and then the embedding in Equation (Equation 1) is performed.

In the watermark extraction stage, according to the watermarked signal y and the pseudorandom sequence p, we can calculate
(2)$$\begin{array}{cc} R= & \frac{{\mathrm{yp}}^{T}}{{\mathrm{pp}}^{T}}\\ = & \frac{\left(x+\alpha \omega p\right)p^{T}}{{\mathrm{pp}}^{T}}\\ = & \alpha \omega +\frac{{\mathrm{xp}}^{T}}{{\mathrm{pp}}^{T}} \end{array}$$
where the superscript ${}^{T}$ denotes the transpose operator. Equation (Equation 2) can be rewritten as
(3)$$R=\left\{\begin{array}{cc} \alpha +\frac{{\mathrm{xp}}^{T}}{{\mathrm{pp}}^{T}},\;\; & \omega =+1\\ -\alpha +\frac{{\mathrm{xp}}^{T}}{{\mathrm{pp}}^{T}},\;\; & \omega =-1 \end{array}\right.$$
where $\frac{{\mathrm{xp}}^{T}}{{\mathrm{pp}}^{T}}$ is referred to as the host signal interference (HSI) since it affects the extraction results and is related to the host signal. If p is long enough to have $\frac{{\mathrm{xp}}^{T}}{{\mathrm{pp}}^{T}}\approx 0$, or if the embedding strength is large enough to have $|\frac{{\mathrm{xp}}^{T}}{{\mathrm{pp}}^{T}}|\ll \alpha$, then *R* has the same sign as ω. Therefore, the embedded watermark bits can be extracted from *R* by
(4)$$\hat{\omega }=\mathrm{sign}\left(R\right)$$

### Channel Model and Embedding Capacity

As defined by information theory [33], a communication channel is a system consisting of an input codebook $I=\{I_{i}\}$, output codebook $O=\{O_{i}\}$ and a probability transition matrix $T=\{p_{ij}\}$, where an element $p_{ij}$ expresses the probability of observing the output $O_{j}$ given the input $I_{i}$. The Figure 1a gives a schematic of a general communication channel with an input codebook size |I| of *m*, an output codebook |O| of *n* and a transition matrix T.

The channel capacity is the highest rate in bits that can be transmitted per channel use and the maximum mutual information between input and output for different input distributions. It can be expressed formulaically as [33],
(5)$$C=\max_{p(I)}I\left(I;O\right)$$

Specifically, if the channel in Figure 1a is weakly symmetric and the input codebook fit a uniform distribution, then its channel capacity is,
(6)$$C=\log\left|O\right|-H\left(p_{i1},p_{i2},\cdots ,p_{in}\right)$$
and $H(p_{i1},p_{i2},\cdots ,p_{in})$ is the Shannon entropy of any row in transition matrix *T*,
(7)$$H\left(p_{i1},p_{i2},\cdots ,p_{in}\right)=-\sum_{j=1}^{n}p_{ij}\log p_{ij}$$
the logarithmic function is set to a base of 2.

According to the above introduction, the VSS watermarking algorithm can be considered a binary symmetric channel, as shown in Figure 1b, in which the input symbols are complemented with probability $P_{e}$, and its channel capacity is
(8)$$C_{vss}=1-H\left(P_{e},1-P_{e}\right)$$
It can be seen that the size of the output codebook corresponding to the VSS algorithm is 2, which is a limitation on the channel capacity. In addition, the error probability of the watermark extraction also dramatically influences the channel capacity. At the extreme, the channel capacity is 0 if the error probability is 0.5; the channel can deliver the message “accurately” if the error probability is dramatically equal to 1.

However, there is a slight difference between channel capacity and embedded capacity. It can be seen from Equation (Equation 1) that embedding a symbol (watermark bit) requires a signal vector of length $l_{x}$. Therefore, the watermarking embedding capacity also needs to take the length of the host signal into account. We define the embedding capacity $E_{c}$ as the bits that can be embedded per sample point, measured in *bits/sample*. It can be calculated from the channel capacity *C*,
(9)$$\begin{array}{cc} E_{c}= & c\cdot \frac{C}{l_{x}}\\ = & c\cdot \frac{\log\left|O\right|-H\left(p_{i1},p_{i2},\cdots ,p_{in}\right)}{l_{x}} \end{array}$$
where *c* is a constant factor representing the loss of capacity in the watermarking system.

In Equation (Equation 9), it can be found that the embedding capacity $E_{c}$ of the watermark is affected by several factors:The output codebook size |O|; in a watermarking system, the input and output codebooks are generally the same in size;The entropy of transition probability $H(p_{i1},p_{i2},\cdots ,p_{in})$, which is relevant to watermark robustness;The length of the signal vector employed to embed a symbol $l_{x}$;The domain transformation and location selection modules contained in the watermarking system. Parameter *c* in Equation (Equation 9) has varied values for different systems.

However, these factors may be mutually constraining, e.g., a short signal vector may heighten the embedding capacity and increase the transition probability’s entropy. Some factors may also conflict with other performance criteria; for instance, a sufficiently large embedding strength reduces the transition probability’s entropy, but it also destroys the imperceptibility of the watermark. The remainder of this paper does not involve the embedding location selection and domain transformation, so the constant factor *c* in Equation (Equation 9) is ignored.

In the next two sections, we present ECSS and DR to improve the two terms corresponding to the numerator in Equation (Equation 9), respectively.

## 4. Extended-Codebook Spread-Spectrum Watermarking Algorithm

The VSS watermarking algorithm adds the pseudorandom sequence p to the signal vector x with different signs according to the watermark bit ω. It has a codebook size of just 2. Hence, increasing the codebook size of the SS-based algorithm is a viable option.

There are *m* mutually orthogonal sequences ${\left\{p_{i}\right\}}_{i=0}^{m-1}$ and the sequences’ norm are 1, i.e.,
(10)$$p_{i}p_{j}^{T}=\left\{\begin{array}{cc} 1,\;\;\;\; & i=j\\ 0,\;\;\;\; & i\neq j \end{array}\right.$$

Here, the length of sequence $p_{i}$ is equal to the signal vector as $l_{x}$ and $l_{x}\geq m$. Each sequence can represent two symbols of the channel input (refer to Equation (Equation 1), a pseudorandom sequence has two signs), then the size of the codebook is extended to |O|=2m. In this section, we propose an extended-codebook spread-spectrum audio watermarking algorithm using the mentioned sequences ${\left\{p_{i}\right\}}_{i=0}^{m-1}$ and analyze its error probability and embedding capacity.

### 4.1. Embedding Approach

When using the sequences ${\left\{p_{i}\right\}}_{i=0}^{m-1}$ as the extended codebook to embed the watermark, we can embed M=log|2m| bits of the watermark once rather than embedding 1 bit in the VSS algorithm, where *m* is assumed to be an integer power of 2. First, the former M−1 bits in the *M*-bit watermark are converted to decimal,
(11)$$k=\sum_{i=0}^{M-2}b_{i}\times 2^{i}$$
where $b_{i}$ is the *i*th bit in the *M*-bit watermark. There are 0≤k≤m−1, so each *M*-bit can correspond to a sequence $p_{k}$. Then, the last bit of the *M*-bit watermark $b_{M-1}$ is employed to modulate the sign of the sequence $p_{k}$. The embedding approach is as follows,
(12)$$y=x-{\left(-1\right)}^{b_{M-1}}\alpha p_{k}$$
The embedded *M*-bit watermark can be expressed as $\left(b_{M-1},k\right)$.

### 4.2. Extraction Approach and Error Probability

For watermark extraction, similar to Equation (Equation 2), a pre-extraction is performed on all sequences,
(13)$$\begin{array}{cc} R_{i}= & {\mathrm{yp}}_{i}^{T}\\ = & {\mathrm{xp}}_{i}^{T}-{\left(-1\right)}^{b_{M-1}}\alpha p_{k}p_{i}^{T}\\ = & \left\{\begin{array}{cc} {\mathrm{xp}}_{i}^{T},\;\;\;\; & i\neq k\\ {\mathrm{xp}}_{k}^{T}-\alpha ,\;\;\;\; & i=k\;\&\;b_{M-1}=0\\ {\mathrm{xp}}_{k}^{T}+\alpha ,\;\;\;\; & i=k\;\&\;b_{M-1}=1 \end{array}\right. \end{array}$$

The HSI term also exists in Equation (Equation 13). When the distribution of the HSI is studied, a Gaussian distribution [9] with mean 0 is usually assumed, and we use Q-Q (quantile–quantile) plot and K-S (Kolmogorov–Smirnov) tests [34] to test this hypothesis on the distribution of the HSI. In the experiments, 5000 samples of the HSI were collected for each length of signal vectors of the VSS watermarking algorithm, and the corresponding assumption was verified using the Q-Q plot and *p*-values of the K-S test.

Figure 2 shows a Q-Q plot and the results of the K-S tests. In Figure 2a, the points are evenly distributed around the reference line, indicating that their distribution is similar to the hypothetical distribution. Due to size limitation, the result is shown here only for the length of the signal vector of 32. In Figure 2b, it is shown that the K-S tests performed on different lengths of signal vector have *p*-values greater than 0.05, which means that we cannot reject the hypothesis that these sample points fit the Gaussian distribution.

We also estimated the HSI’s mean and standard deviation, as shown in Figure 3. The mean of the HSI can also be considered as 0. The standard deviation σ is insignificantly related to the length of the signal vector.

When the length of $p_{i}$ is appropriate, or the embedding strength is large, the *k* corresponding to the watermark can be obtained by comparing the magnitudes of the pre-extraction, i.e.,
(14)$$\hat{k}=\underset{i\in \{0,1,\cdots ,2^{M-1}\}}{\mathrm{argmax}}\left|R_{i}\right|$$
$b_{M-1}$ is obtained from the sign of $R_{\hat{k}}$,
(15)$${\hat{b}}_{M-1}=\left\{\begin{array}{cc} 0,\; & R_{\hat{k}}<0\\ 1,\; & R_{\hat{k}}\geq 0 \end{array}\right.$$
Thus, the *M*-bit watermark is extracted as $\left({\hat{b}}_{M-1},\hat{k}\right)$. However, the extracted watermark may still be incorrect due to the HSI.

Generally, the embedded watermark of $\left(1,k_{0}\right)$ is analyzed here as an example. The error occurs in two cases, i.e., $\hat{k}\neq k_{0}$ in Equation (Equation 14), or $\hat{k}=k_{0}$, but $b_{M-1}\neq 1$ in Equation (Equation 15).

When $\hat{k}\neq k_{0}$, there is at least one pre-extraction $R_{m}$ greater in magnitude than $R_{k_{0}}$. It can be described as,
(16)$$\left|R_{m}\right|\geq \left|R_{k_{0}}\right|$$

Equation (Equation 16) can be analyzed in two scenarios. The first scenario is
(17)$$R_{m}\geq {\mathrm{xp}}_{k_{0}}^{T}+\alpha$$
In this case, $R_{m}=\max\left(R_{0},\cdots ,R_{k_{0}-1},R_{k_{0}+1},\cdots ,R_{m-1}\right)$, and all the pre-extractions are independent and identically distributed. Then, the cumulative density function (CDF) of $R_{m}$ is,
(18)$$F_{m}\left(x;\sigma \right)=\Phi^{m-1}\left(x;\sigma \right)$$
where Φ(x;σ) is the CDF of the HSI.

The right side of Equation (Equation 17) is a rather tricky variable to analyze, and since the expectation of the HSI ${\mathrm{xp}}_{k_{0}}^{T}$ is 0, Equation (Equation 17) is simplified here as follows,
(19)$$\max(R_{0},\cdots ,R_{k_{0}-1},R_{k_{0}+1},\cdots ,R_{m-1})\geq \alpha$$
The probability that Equation (Equation 19) holds is
(20)$$p_{e1}=1-\Phi^{m-1}\left(\alpha ;\sigma \right)$$

Similarly, the second scenario of Equation (Equation 16) is,
(21)$$R_{m}\leq {\mathrm{xp}}_{k_{0}}^{T}-\alpha$$
The corresponding error probability is
(22)$$p_{e2}=1-{\left(1-\Phi \left(-\alpha ;\sigma \right)\right)}^{m-1}$$

Since the probability density function (PDF) of the HSI in Equation (Equation 3) is even, we have $p_{e1}=p_{e2}$. Therefore, the probability of the first case of error (Equation (Equation 16)) is
(23)$$\begin{array}{cc} p_{e}= & 1-\left(1-p_{e1}\right)\left(1-p_{e2}\right)\\ = & 1-{\left(1-p_{e1}\right)}^{2}\\ = & 1-\Phi^{2m-2}\left(\alpha ;\sigma \right) \end{array}$$

When $\hat{k}=k_{0}$ but ${\hat{b}}_{M-1}\neq 1$, there is
(24)$${\mathrm{xp}}_{k_{0}}^{T}\leq -2\alpha$$

It is noted that Equation (Equation 16) does not hold at this point, so the corresponding probability is
(25)$$\begin{array}{cc} p_{e3}= & \left(1-p_{e}\right)\Phi \left(-2\alpha ;\sigma \right)\\ = & {\left(1-p_{e1}\right)}^{2}\Phi \left(-2\alpha ;\sigma \right)\\ = & \Phi^{2m-2}\left(\alpha ;\sigma \right)\Phi \left(-2\alpha ;\sigma \right) \end{array}$$

In a nutshell, the probability that Equations (Equation 14) and (Equation 15) can extract the correct watermark is
(26)$$\begin{array}{cc} p_{c}= & 1-p_{e}-p_{e3}\\ = & \Phi^{2m-2}\left(\alpha ;\sigma \right)-\Phi^{2m-2}\left(\alpha ;\sigma \right)\Phi \left(-2\alpha ;\sigma \right)\\ = & \Phi^{2m-2}\left(\alpha ;\sigma \right)\Phi \left(2\alpha ;\sigma \right) \end{array}$$
Since the distribution of the HSI is symmetric, the mentioned probabilities still hold at $b_{M-1}=0$, and they are all functions of the length of the signal vector $l_{x}$, the number of sequences *m* and the embedding strength α.

### 4.3. Embedding Capacity

Two error cases were analyzed in Section 4.2, where the error probability $p_{e}$ was uniformly distributed to 2m−2 corresponding symbols, while $p_{e3}$ only corresponded to one symbol. Therefore, one row of the transition probability matrix can be written as,
(27)$$r=\left(\frac{p_{e}}{2m-2},\frac{p_{e}}{2m-2},\cdots ,p_{e3},p_{c}\right)$$
where there are a total of (2m−2) terms $\frac{p_{e}}{2m-2}$.

The embedding capacity of the ECSS algorithm can be calculated according to Equation (Equation 9),
(28)$$E_{c}=\frac{\log2m-H(r)}{l_{x}}$$
where the constant factor *c* is ignored here. Equation (Equation 28) shows that $E_{c}$ becomes lower when the signal vector becomes longer.

Since $l_{x}\geq m$, the maximum embedding capacity for a given *m* is obtained when
(29)$$l_{x}=m$$

Thus, the embedding capacity is simplified to a function of the number of sequences (the length of the signal vector) *m* and the embedding strength α.

## 5. Diversity Reception

It can be seen in Equation (Equation 28) that the transition probability of the channel, especially the accuracy of the watermark extraction, $p_{c}$, has a significant influence on the embedding capacity $E_{c}$. Improving the watermark extraction accuracy is also an effective way to heighten the embedding capacity. Moreover, a high accuracy is also beneficial for the implementation of a watermarking system.

There are three strategies to improve the accuracy of the watermark extraction shown in Equation (Equation 26).

(1)Reducing the number of sequences *m*;(2)Increasing the embedding strength α;(3)Modifying the distribution of the HSI.

Both strategies (1) and (2) involve a parameter selection, and in the next section, we experiment with different parameters.

In this section, we adopt the diversity reception (DR) mechanism to modify the HSI distribution, a method commonly used in signal detection. The DR mechanism can effectively improve the accuracy of the watermark extraction. In brief, the *M*-bit watermark $\left(b_{M-1},k\right)$ is repeatedly embedded into different signal vectors $x_{\gamma }$, where γ is an integer in the interval [0,Γ), where Γ is the number of repetitions in DR. In the extraction phase, all the watermarked signal vectors are referenced. Equation (Equation 13) can be rewritten as,
(30)$$\begin{array}{cc} R_{i}= & \frac{1}{\Gamma }\left(\sum_{\gamma =0}^{\Gamma -1}y_{\gamma }\right)p_{i}^{T}\\ = & \frac{1}{\Gamma }\left(\sum_{\gamma =0}^{\Gamma -1}\left(x_{\gamma }-{\left(-1\right)}^{b_{M-1}}\alpha p_{k}\right)\right)p_{i}^{T}\\ = & \left\{\begin{array}{cc} \frac{1}{\Gamma }\left(\sum_{\gamma =0}^{\Gamma -1}x_{\gamma }p_{i}^{T}\right),\;\;\;\; & i\neq k\\ \frac{1}{\Gamma }\left(\sum_{\gamma =0}^{\Gamma -1}x_{\gamma }p_{k}^{T}\right)+\alpha ,\;\;\;\; & i=k\;\;\&\;\;b_{M-1}=1\\ \frac{1}{\Gamma }\left(\sum_{\gamma =0}^{\Gamma -1}x_{\gamma }p_{k}^{T}\right)-\alpha ,\;\;\;\; & i=k\;\;\&\;\;b_{M-1}=0 \end{array}\right. \end{array}$$

By substituting the pre-extractions of Equation (Equation 30) into Equations (Equation 14) and (Equation 15), the corresponding embedded watermark can be extracted. Since $x_{\gamma }p_{i}^{T}$ or $x_{\gamma }p_{k}^{T}$ are Γ independent and identically distributed random variables with variance $\sigma^{2}$, the variance of $\frac{1}{\Gamma }\left(\sum_{\gamma =0}^{\Gamma -1}x_{\gamma }p_{i}^{T}\right)$ or $\frac{1}{\Gamma }\left(\sum_{\gamma =0}^{\Gamma -1}x_{\gamma }p_{k}^{T}\right)$ is $\frac{\sigma^{2}}{\Gamma }$. Hence, in the DR mechanism, the distribution of the HSI is modified, and the variance is smaller than that in the ECSS algorithm. It is indicated from Equation (Equation 26) that the smaller the standard deviation of the HSI is, the higher the accuracy of the watermark extraction is.

This method of repeatedly embedding the watermark and averaging the HSI in different watermarked signal vectors to reduce the randomness of the HSI is referred to as DR in this paper. Although DR improves the accuracy of the watermark extraction, on the other hand, DR may also reduce the embedding capacity because more signal vectors are used for embedding the watermark.

It seems that DR is a trade-off to balance the embedding capacity with the robustness. However, it is not the case. Even if the number of repetitions is small (e.g., Γ=2), DR can still significantly improve the accuracy of the watermark extraction (in Section 7.3). In this case, the gain of an accuracy improvement on the embedding capacity outweighs the capacity cost of repeated embeddings.

## 6. Our Audio Watermarking System

An audio-watermarking-system-based ECSS and DR is provided here. It can be divided into two stages, the watermark embedding stage and the extraction stage, as shown in Figure 4.

### 6.1. The Embedding Stage

The watermark embedding stage is shown as Figure 4a and can be summarized as three steps.


*Step 1: framing, DCT and selection of embedding regions*


The host audio is split into consecutive nonoverlapping frames and DCT is applied on each frame; the medium frequency components of the DCT coefficients are selected as the embedding regions and grouped as vectors $x_{\gamma }$.


*Step 2: watermarking modulation and embedding*


The sequences ${\left\{p_{i}\right\}}_{i=0}^{m-1}$ is an ensemble of standard orthogonal vectors satisfying Equation (Equation 10) and of length *m*. The *M*-bit watermark is represented in the form of $\left(b_{M-1},k\right)$ and repeatedly embedded into multiple signal vectors $x_{\gamma }$ according to Equation (Equation 12), resulting in watermarked vectors $y_{\gamma }$. The detail of this step is described in Section 4.1.


*Step 3: audio reconstruction*


Replace $x_{\gamma }$ correspondingly by the watermarked vector $y_{\gamma }$. The DCT coefficients are reorganized, and then an inverse DCT is performed on the coefficients to obtain the audio frames. These audio frames are concatenated to yield the watermarked audio.

### 6.2. The Extraction Stage

The watermark extraction stage is shown in Figure 4b and can also be summarized as three steps.


*Step 1: framing, DCT and the selection of embedding regions*


The received audio is split into consecutive nonoverlapping frames and DCT is applied on each frame; the medium frequency components of the DCT coefficients are selected as the embedding regions and grouped as vectors $y_{\gamma }$.


*Step 2: averaging and pre-extraction*


Average the $y_{g}amma$s yielding $\bar{y}$, whose correlation with each sequence $p_{i}$ is the pre-extraction results $R_{i}$’s, as Equation (Equation 30) in Section 5.


*Step 3: extraction*


The watermark $\left({\hat{b}}_{M-1},\hat{k}\right)$ can be obtained from the magnitude and sign of the pre-extraction results, as shown in Equations (Equation 14) and (Equation 15) of Section 4.2.

The ECSS and DR correspond to Step 2 in the embedding stage and Steps 2 and 3 in the extraction stage of our watermarking system. Framing, DCT, and the selection of embedding regions can be replaced by any embedding regions’ strategy. The watermarking system presented in this section is only used as an example to evaluate the performances of the ECSS and DR.

## 7. Experiments and Discussion

This section uses a series of experiments to validate the proposed algorithm.

The accuracy of the watermark extraction of the ECSS/+DR algorithm was measured at different embedding strengths and lengths of signal vectors. The embedding capacity was calculated according to the accuracy. The performance of the ECSS/+DR algorithm was also compared with that of the VSS algorithm to validate the effectiveness of the ECSS algorithm.

The influence of the DR mechanism with different repetitions Γ on the performance of the audio watermarking system was also evaluated. This part of the experiment also verified the effectiveness of the DR mechanism.

In addition, we evaluated the robustness of the proposed algorithm against common signal processing attacks and discuss the influence of these attacks on the embedding capacity.

Finally, the performance of our algorithm was also compared with that of other existing watermarking algorithms.

### 7.1. Experimental Settings and Metrics

Our watermarking algorithm contains three settable parameters, namely the embedding strength α, the length of signal vectors *m* (equal to the number of sequences, as in Equation (Equation 29)) and the number of repetitions Γ. We chose different α,m and Γ for the experiments, evaluated the influence of attacks on the performance of the algorithm and compared the performance with other existing algorithms on suitable parameters.

We selected 1000 stereo audio tracks with a sample rate of 44.1 kHz and a duration of 30 s from the FMA dataset [35] for the experiments. In the selection of embedding regions, the framed audio segment had 2048 samples and we used the 256th to 511th DCT coefficients as the embedding region, which corresponded to the frequency interval [2756,5512] Hz. All or part of these 256 DCT coefficients were divided into several signal vectors.

The embedding distortion SWR (signal-to-watermark ratio) was employed to evaluate the watermark imperceptibility. The SWR is calculated as,
(31)$$\mathrm{SWR}=10\mathrm{lg}\left(\frac{\sum_{i=1}^{n}x^{2}\left(i\right)}{\sum_{i=1}^{n}{\left(y(i)-x(i)\right)}^{2}}\right)$$
where x(i) and y(i) denote the *i*th sample of the host audio and the watermarked audio. SWR reflects the amount of change to the host audio by the watermark embedding. Although SWR does not reflect the perceptual quality of audio locally, it is still an important metric of imperceptibility.

In this paper, the robustness of the watermarking algorithm was evaluated using the watermark group extraction precision (GEP). A watermark group consists of the watermark bits corresponding to a sequence, e.g., an *M*-bit watermark in the ECSS algorithm, while a watermark group is a single bit in the VSS algorithm. The GEP is calculated as follows,
(32)$$\mathrm{GEP}=\frac{\#\;of\;groups\;extracted}{\#\;of\;total\;groups}\times 100\%$$
where GEP=1 means that the watermark can be extracted entirely and correctly. The higher the GEP is, the more robust the watermarking algorithm is.

When evaluating the robustness, we also used another metric similar to the GEP, the watermark bit detection rate (BDR). The BDR compares the embedded watermark bits with the extracted bits, and it is defined as,
(33)$$\mathrm{BDR}=\frac{\#\;of\;bits\;extracted}{\#\;of\;total\;bits}\times 100\%$$
where BDR=1 means that the watermark can be extracted entirely and correctly. The higher the BDR is, the more robust the watermarking algorithm is.

We used two different metrics to evaluate robustness, because

(1)The BDR does not reflect the actual robustness of the algorithms if there is a use of multibit embedding, and in the case of the ECSS algorithm, e.g., when the four-bit watermark (1, 7) is incorrectly extracted as (1, 6), the BDR is 75%, while the GEP is 0%. The GEP reflects the fact more accurately.(2)However, not all algorithms embed the same quantity of bits once. For those algorithms that have small watermark groups, the GEP typically yields better results. The actual performance of the algorithms with various embedding bits is somewhat reflected in the BDR.

We used the GEP to evaluate the robustness of the ECSS algorithm and DR, and then used the BDR, which is more common in the literature, to evaluate various attacks and compare them with existing algorithms.

### 7.2. Effectiveness of the ECSS Algorithm

In this subsection, we statistically measure the GEPs of the ECSS algorithm and calculate the corresponding embedding capacity $E_{c}$ based on the GEPs. The relevant performance of the VSS algorithm is also evaluated as a comparison.

The GEP can only estimate the probability of correctly extracting the watermark group, but the complete probability transition matrix is required to calculate the embedding capacity. For this reason, we assigned the error probabilities uniformly to the other symbols to calculate the infimum of the embedding capacity. Figure 5, Figure 6, Figure 7 and Figure 8 show the results of these experiments, where Figure 5 shows the GEP versus embedding strength α at different lengths of the signal vector; Figure 6 shows the embedding capacity versus embedding strength α at different lengths of the signal vector; Figure 7 shows the GEP versus the length of the signal vector at different embedding strengths; Figure 8 shows the embedding capacity versus the length of the signal vector at different embedding strengths.

As shown in Figure 5, the GEP of the VSS algorithm varies flatly and significantly outperforms that of the ECSS algorithm. As the GEP favors the single-bit embedding approaches, the performance difference between the proposed algorithm and the VSS algorithm is not as significant as shown in Figure 5. As the embedding strength increases, the GEP of ECSS gradually increases and finally saturates. In Figure 6, the embedding capacity of VSS is greater than that of the ECSS algorithm when the embedding strength is low. As the embedding strength becomes larger, the embedding capacity of both algorithms increases, but the increase is faster for the ECSS algorithm. It can also be seen that the embedding capacity of VSS is low when the embedding strength is greater than 0.3.

In Figure 7, it can be seen that the GEP of VSS is almost independent of the length of the signal vector. The GEP of VSS increases with the lengthening of the signal vector for a given embedding strength. However, the factor $\sqrt{m}$ was divided by the embedding strength to ensure that the SWR of the VSS and ECSS algorithms were equal. (Although α in both Equations (Equation 1) and (Equation 12) denotes the embedding strength, the influence on the embedding distortion is different since the norm of the pseudorandom sequence in Equation (Equation 1) is $l_{x}$, while it is one in Equation (Equation 12). Thus, there is a factor $\sqrt{l_{x}}$ difference between the embedding strengths corresponding to the same embedding distortion.) At the embedding strength α=0.2 or α=0.3, the GEPs of the ECSS algorithm do not reach saturation, and there is no obvious pattern indicating that the GEP is related to the length of the signal vector, which is similar to the standard deviation of the HSI in Figure 3b.

On the other hand, it is shown in Figure 8 that the embedding capacity decreases as the length of the signal vector grows, which can be concluded accordingly from Equation (Equation 28).

### 7.3. Effectiveness of the DR mechanism

In the above subsection, the GEP and embedding capacity of VSS and ECSS were analyzed. However, the role of the DR mechanism is not yet clear. In this subsection, the performance of ECSS is analyzed with different repetitions of the DR mechanism combined. The relevant experimental results are shown in Figure 9, Figure 10, Figure 11 and Figure 12. Figure 9 and Figure 10 show the curves of GEPs and embedding capacity versus the embedding strength at different lengths of the signal vector, where “w/o DR” means Γ=1. In this paper, the length of the signal vector in the DR mechanism refers to the sum of the length of multiple repeated signal vectors, e.g., the length of 64 for the ECSS+DR algorithm in Figure 9 means that two signal vectors with length 32 were used to embed the same *M*-bit watermark. The DR mechanism with repetition Γ=3 in Figure 9 and Figure 10 actually used signal vectors with lengths 33, 63, 129, 255, which are somewhat different from the subplot titles. Figure 11 and Figure 12 show the curves of GEPs and embedding capacity versus the length of the signal vector at different embedding strengths.

As the embedding strength α increase, the GEP gradually increases and eventually saturates, as shown in Figure 9. Furthermore, the smaller the embedding strength is required to saturate the GEP when there are more repetitions. As in Figure 10, the embedding capacity of the watermark is improved with the embedding strength. However, unlike the GEP, DR with different number of repetitions has different embedding capacities when it finally reaches saturation, and the embedding capacity at saturation is lower when there are more repetitions. At low embedding strength, the ECSS+DR algorithm achieves a better imperceptibility, robustness, and embedding capacity than the ECSS algorithm. Moreover, as the signal vector lengthens, the embedding capacity with different repetitions gets closer when saturated, which indicates that repeated embedding is less costly.

There is no clear relationship between the GEP and the length of the signal vector, as shown in Figure 11. Nevertheless, it can be seen in Figure 12 that the embedding capacity gradually decreases as the length of the vector gets longer. If the chosen embedding strength saturates the GEP, the embedding capacity is inversely proportional to the square root of the length of the signal vector.

In summary, the DR mechanism effectively improves the embedding capacity and the GEP at low embedding strength, balancing the proposed watermarking algorithm’s imperceptibility, robustness and embedding capacity.

### 7.4. Robustness and Embedding Capacity against Attacks

In this subsection, we evaluate the BDR and embedding capacity of the proposed ECSS and ECSS+DR algorithms for some common data processing attacks. The attacks are described in Table 1.

In order to demonstrate the influence of the attacks on the performance of the watermarking algorithm, the parameters α=0.3 and Γ=2 for which the GEP did not reach saturation were deliberately chosen. The evaluation was also performed using signal vectors of different lengths. The experimental results are shown in Figure 13, the top subplot is the BDR and the bottom half is the corresponding embedding capacity.

The effect of the signal processing attacks can be equivalent to adding a certain amount of noise to the audio, and the power of that noise is much lower than the power of the audio itself. The ECSS algorithm proposed in this paper resisted most of these attacks, where there was almost no influence in RQZ, ASC, RSP, LPF, and MP3 attacks, which reduced the average BDR of the ECSS algorithm by 0.1% to 0.5% and had negligible influence on the ECSS+DR algorithm. However, in the WGN attack, the BDR of the ECSS algorithm showed a substantial decline, where a 15 dB Gaussian noise would decrease the BDR of the ECSS algorithm by 4.1%, corresponding to a 1.0% decrease for the ECSS+DR algorithm, and a 10 dB noise would decrease the BDR of the ECSS algorithm by 11.5%, corresponding to a 5.5% decrease for the ECSS+DR algorithm.

The decrease of the BDR also reduced the embedding capacity of algorithms. Among these attacks, the influence of RQZ, ASC, RSP, LPF, and MP3 attacks on the embedding capacity of the watermarking algorithms was less than $3\times 10^{-5}$ bits/sample, which is negligible. The WGN attack decreased the embedding capacity of the ECSS and ECSS+DR algorithms by $5\times 10^{-4}$ bits/sample and $3\times 10^{-4}$ bits/sample, respectively, at an SNR of 15 dB; 10dB noise decreases the embedding capacity by $1.2\times 10^{-3}$ bits/sample and $1.5\times 10^{-3}$ bits/sample correspondingly.

We also evaluated the performance of the ECSS/+DR algorithm against a short delay attack (DLY). The BDR was reduced by 8.11% for the ECSS algorithm and by 2.51% for the ECSS+DR algorithm. Due to the change in robustness, the embedding capacity was reduced by $9.1\times 10^{-4}$ and $7.5\times 10^{-4}$, respectively. Algorithm DLY is not a signal processing attack, although adding other parts from the same audio is still equivalent to adding noise. Indeed, the ECSS+DR algorithm provided an anti-interference property.

The influence of various attacks on the performance and the equivalent noise intensity are summarized in Table 2, where the equivalent SNR indicates the intensity of the equivalent noise corresponding to the attack.

In addition to signal processing attacks, advanced attacks such as crop, framing errors and playback scaling are also possible with watermarked audio. It is difficult to resist these attacks by considering only the embedding and extraction algorithms. Various strategies for selecting embedding regions have been proposed in the literature, among which RASE [36] and RFPS [37] can resist crop and framing errors, A WSOLA-based strategy [38] can resist playback. These strategies can replace Step 1 in Section 6 to resist the corresponding attacks. However, there is no watermarking algorithm that can resist all audio attacks.

### 7.5. Comparison with Other Existing Algorithms

The proposed ECSS+DR algorithm was also compared with existing high-embedding-capacity algorithms [12,13,15,16]. These algorithms all have relatively high embedding capacity which have been presented as related works in Section 2, and the parameter settings of each comparison algorithm are referred to in the relevant papers. Here, the BDR was used to evaluate the robustness of these algorithms against various attacks.

It is still evident from the experimental results in Table 3 that our algorithm obtained outstanding performance. “Ours” represents the ECSS+DR algorithm, where the embedding strength was α=0.35, the repetitions Γ=2, and the length of the signal vector m=16 (due to repeated embeddings, each *M*-bit watermark group actually used signal vectors with a total length of 32). These parameters provide a balanced performance for the ECSS+DR algorithm according to Section 7.2 and Section 7.3.

The ECSS+DR algorithm had similar embedding distortion SWR as the algorithm in [12] and also had a comparable BDR in CLP, LPF, and RSP attacks. However, the embedding capacity of our algorithm was significantly higher, and our algorithm was also more robust against WGN and MP3 attacks. Compared with the algorithm in [15], the robustness of our algorithm was close to that of the latter, but our embedding capacity was better, and notably, there was a slighter SWR in our algorithm. Compared with the methods in [13,16], our embedding distortion was near to theirs, but our algorithm performed much better in both robustness and embedding capacity.

In a nutshell, our algorithm achieved a significant advantage in embedding capacity, without any disadvantage in robustness and imperceptibility compared to other existing algorithms.

## 8. Summary

In this paper, we analyzed the spread-spectrum watermarking algorithm from the perspective of the communication channel and formularized its watermark embedding capacity. According to this analysis, we proposed a watermarking algorithm using extended-codebook ECSS and optimized it by adopting the diversity reception (DR) mechanism, so that the ECSS+DR watermarking algorithm achieved a higher embedding capacity and stronger robustness while given a limited embedding strength.

However, the embedding capacity formulation in this paper was only limited to the spread-spectrum watermarking algorithm, and the derived formula did not take into account the various attacks that could be encountered with the watermarked audio. Due to the excellent anti-interference performance of the spread-spectrum approach, our algorithm still resisted common signal processing attacks. Our team will subsequently try to build a more generalized embedding capacity model that takes into account watermarking algorithms from other paradigms such as QIM and deep learning, as well as considering attacks.

In this paper, a Gaussian distribution was used to model the HSI, which is a heuristic method lacking a rigorous theoretical support. Although the Gaussian distribution passed the K-S test, we note that the sample points in the Q-Q plots formed an S-like curve. Therefore, there may exist a more suitable distribution for modeling the HSI.

Our algorithms are not resistant to white-box attacks in which the attackers knows this algorithm. Although the embedder and the extractor of the watermarking system are the same, the security of this algorithm still needs to be improved in future work.

## Figures and Tables

**Figure 1 entropy-24-01843-f001:**
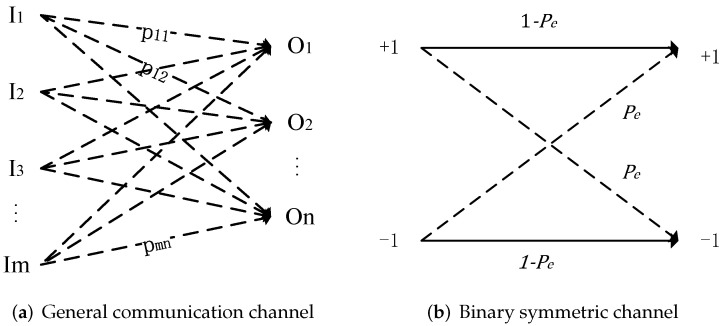
Schematic of two types of channel.

**Figure 2 entropy-24-01843-f002:**
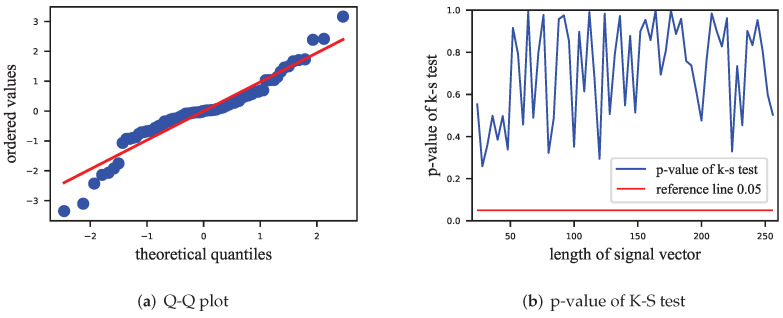
Distribution test of HSI for ECSS. (**a**) Q-Q plot at signal length of 32, with most points near the red line; (**b**) *p*-values of K-S test, with *p*-values greater than 0.05 at all signal lengths.

**Figure 3 entropy-24-01843-f003:**
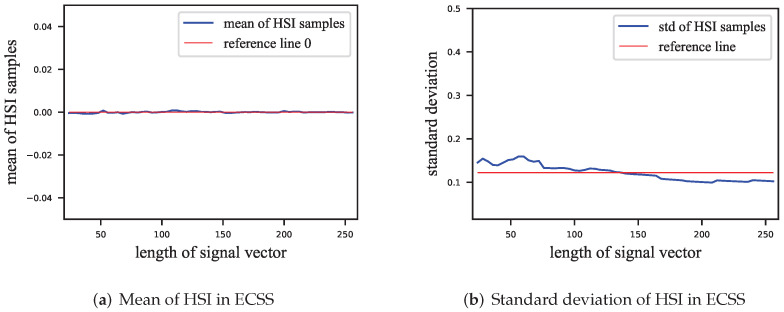
Estimation of the mean and standard deviation of the HSI of the ECSS. (**a**) Mean of HSI, close to 0; (**b**) standard deviation of HSI. The longer the signal vector is, the smaller the standard deviation of HSI is, but this relationship is insignificant.

**Figure 4 entropy-24-01843-f004:**
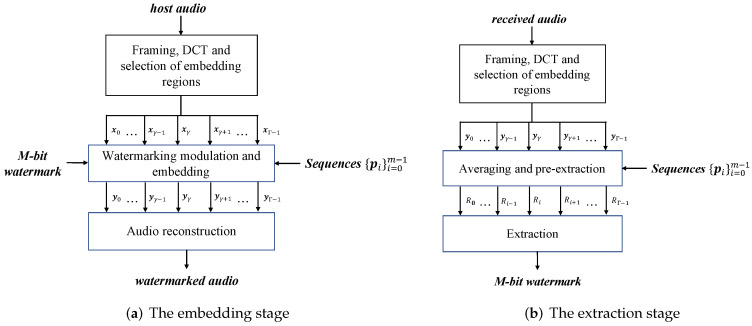
Sketch of our audio watermarking system.

**Figure 5 entropy-24-01843-f005:**
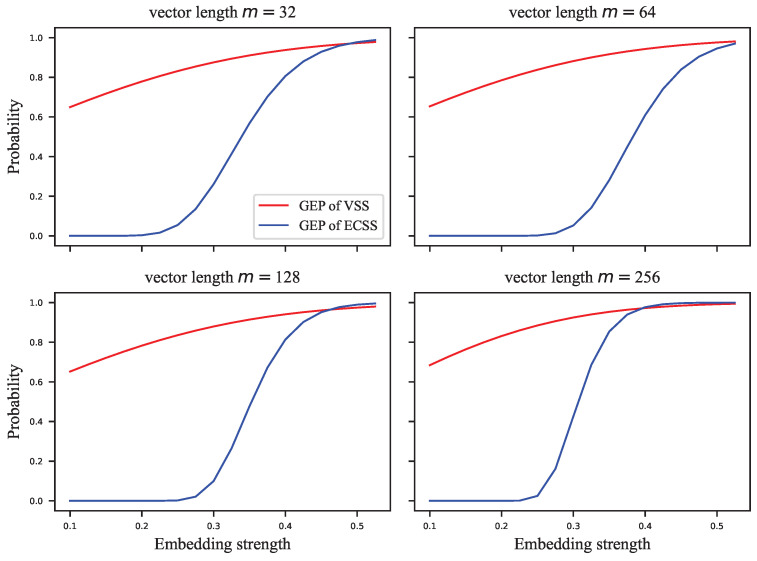
The GEP versus embedding strength at different lengths of signal vector.

**Figure 6 entropy-24-01843-f006:**
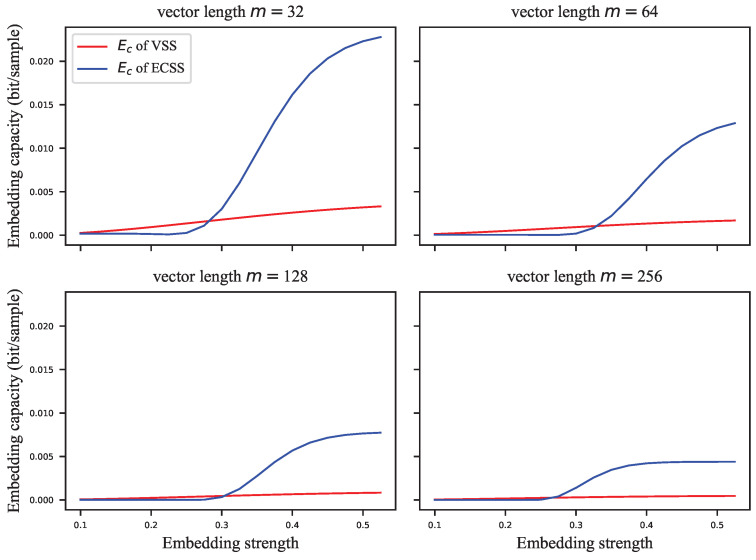
The embedding capacity versus embedding strength at different lengths of signal vector.

**Figure 7 entropy-24-01843-f007:**
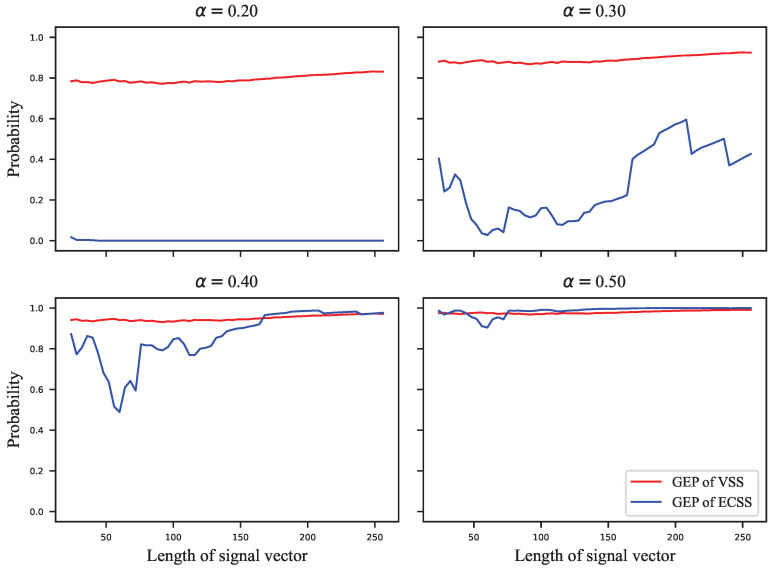
The GEP versus length of signal vector at different embedding strengths.

**Figure 8 entropy-24-01843-f008:**
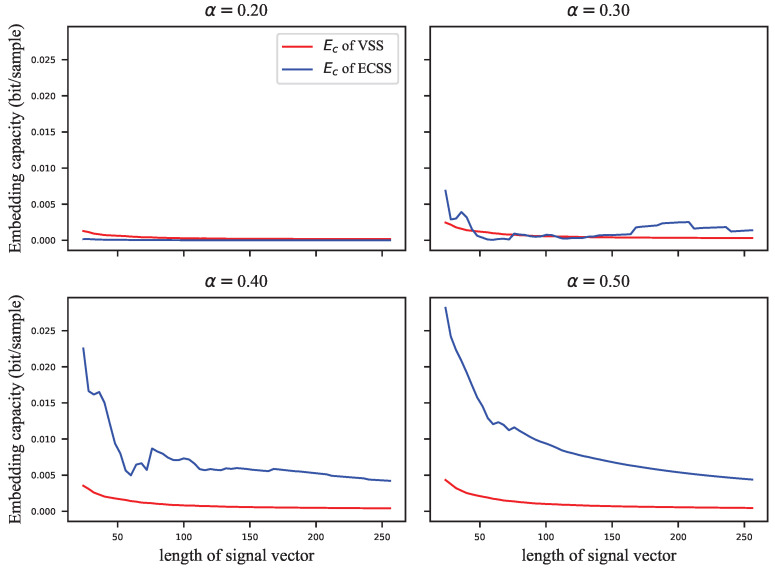
The embedding capacity versus length of signal vector at different embedding strengths.

**Figure 9 entropy-24-01843-f009:**
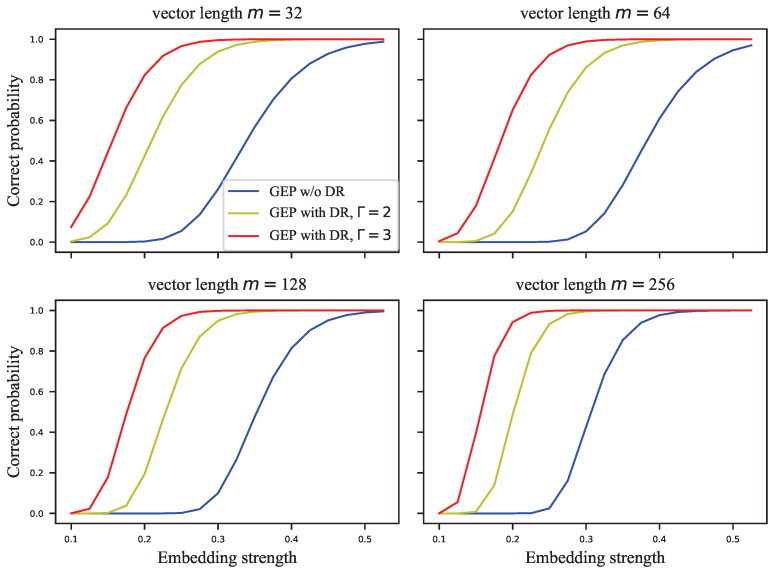
The GEP versus embedding strength at different lengths of signal vector.

**Figure 10 entropy-24-01843-f010:**
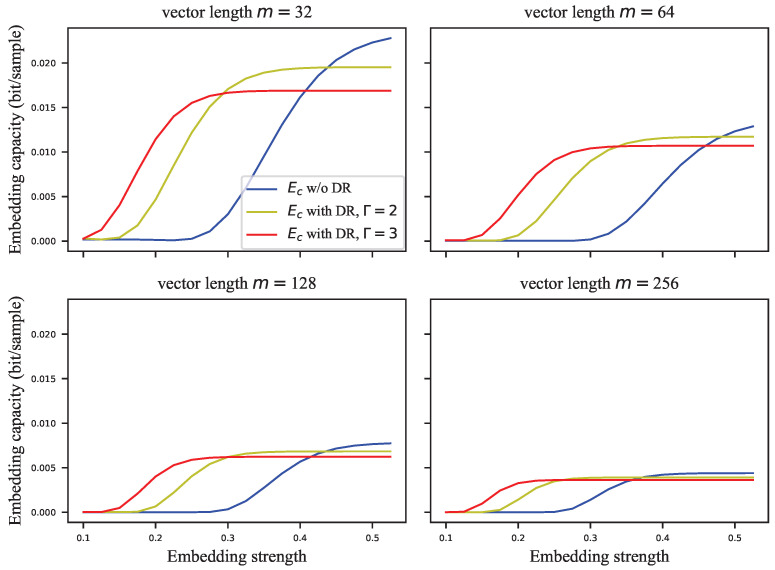
The embedding capacity versus embedding strength at different lengths of signal vector.

**Figure 11 entropy-24-01843-f011:**
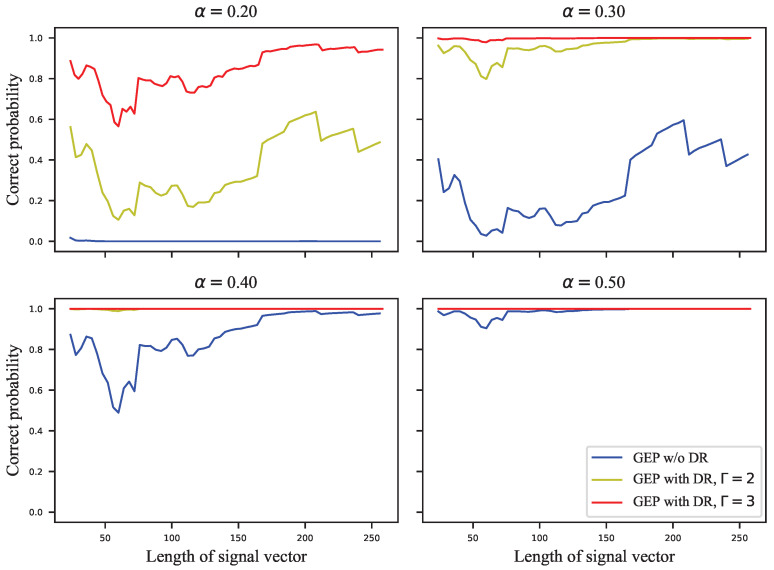
The GEP versus length of signal vector at different embedding strengths.

**Figure 12 entropy-24-01843-f012:**
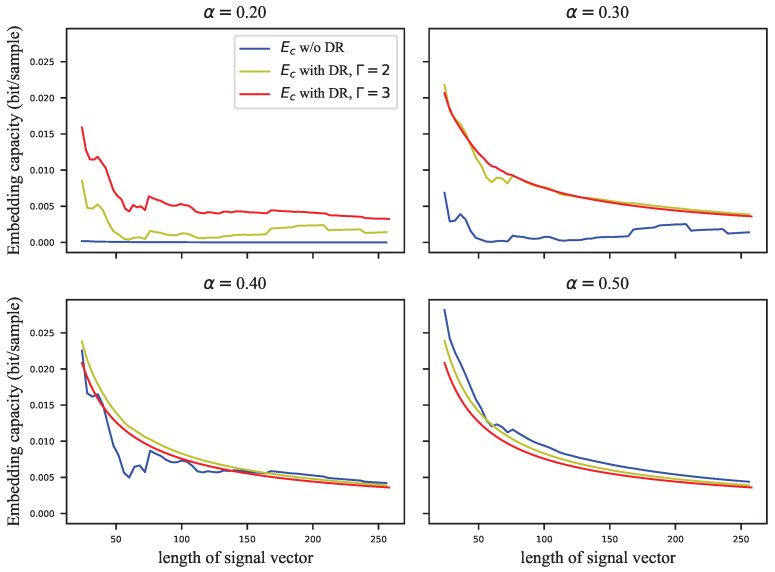
The embedding capacity versus length of signal vector at different embedding strengths.

**Figure 13 entropy-24-01843-f013:**
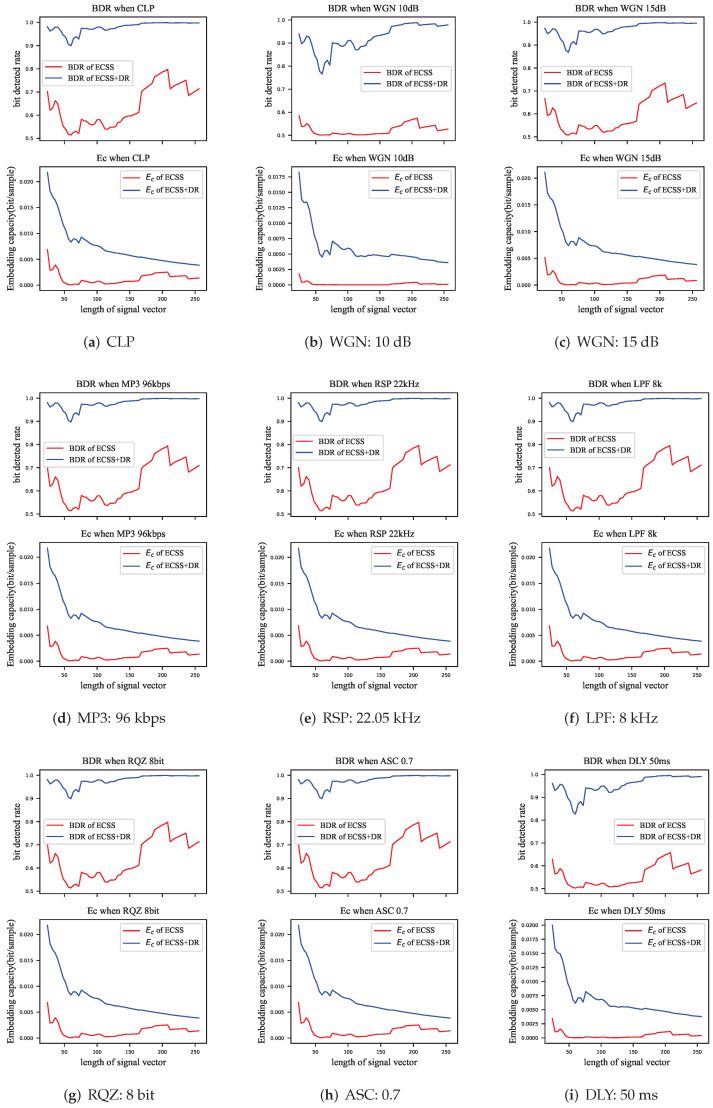
The BDR (top) and the embedding capacity (bottom) of the algorithm against attacks, The horizontal coordinates indicate the different lengths of the signal vector. Red: ECSS, blue:ECSS+DR.

**Table 1 entropy-24-01843-t001:** The attacks used to evaluate robustness and embedding capacity.

Abbrev.	Attacks	Description
CLP	Closed loop	Watermarked audio is in fact not attacked.
WGN	Adding white Gaussian noise	Gaussian noise with SNR of 15 dB or 10 dB is added.
MP3	MP3 compression	MPEG-1 Layer-III compression is applied to water-
		marked audio with a bitrate of 96 kbps.
RSP	Resampling	Watermarked audio downsampled to 22.05 kHz,
		then upsampled to 44.1 kHz.
LPF	Lowpass filter	Watermarked audio is filtered by a Butterworth
		filter, with an 8 kHz cutoff frequency.
ASC	Amplitude scaling	The amplitude of watermarked audio is diminished
		by 0.7 times.
RQZ	Requantization	The 16-bit watermarked audio is requantized down
		to 8 bits/sample and back to 16 bits/sample.
DLY	Short delay	A 50 ms delayed copy of watermarked audio is
		added with an intensity of 0.3

**Table 2 entropy-24-01843-t002:** The average influence of attacks on the watermarking algorithm.

Attacks	Avg. Degraded BDR (%)		Avg. Degraded $E_{c}$ (Bits/Sample)		Equivalent SNR (dB)
	ECSS	ECSS+DR	ECSS	ECSS+DR	
CLP	0	0	0	0	-
WGN 10 dB	11.5	5.5	$1.2\times 10^{-3}$	$1.5\times 10^{-3}$	10
WGN 15 dB	4.1	1.0	$5\times 10^{-4}$	$3\times 10^{-4}$	15
MP3	0.3	0.1	$3.5\times 10^{-5}$	$1.7\times 10^{-5}$	30
RSP	0.1	0.05	$1.1\times 10^{-5}$	$5.5\times 10^{-6}$	35
LPF	0.1	0.01	$1.7\times 10^{-5}$	$8.3\times 10^{-6}$	33
RQZ	0	0	0	0	45
ASC	0	0	0	0	55
DLY	8.11	2.51	$9.1\times 10^{-4}$	$7.5\times 10^{-4}$	13

**Table 3 entropy-24-01843-t003:** The performance of ECSS+DR and existing algorithms.

Algorithm		Hwang [12]	Wang [13]	Xiang [15]	Zhang [16]	Ours
Metrics						
$E_{c}$ (bits/sample)		0.0039	0.0011	0.0019	0.0015	**0.0189**
SWR (dB)		25	28.42	22.74	25.63	24.55
CLP		99.67	99.64	100	80.59	99.38
WGN 10dB		55.42	84.72	99.3	74.83	95.10
LPF		98.16	88.63	100	60.08	99.37
MP3		70.29	89.52	99.8	65.01	99.36
RSP		99.36	97.66	97.7	-	99.37

## Data Availability

Not applicable.

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
