# Peer review of "ECSS: High-Embedding-Capacity Audio Watermarking with Diversity Reception"

_entropy, 2022, doi:10.3390/e24121843_

Round 1

Reviewer 1 Report

The authors propose the extended codebook spread spectrum (ECSS) watermarking algorithm to heighten embedding capacity. Overall, the topic of this paper is convincing and the problem is hot.

This paper is very original and very solid in both theoretical study and experiments. It can definitely publish in an MDPI journal. 

Reviewer 2 Report

The paper deals with a relevant problem.

The proposed watermarking algorithm sounds good.

However, it is another application of SS wm algorithm .... so the basic ideas characterizing the proposed algorithm are well-known in the literature...

Some improvements:

-some formulas in Section 4 have to be better explained

-some parameters set in Section 7 have to be better explained....

Reviewer 3 Report

The authors state: “In this paper, the spread spectrum watermarking algorithm is viewed as a communication channel, and the embedding capacity is analyzed and modeled with information theory.”. This is a theoretical topic that should be presented itself in one separate paper. It is confusing to have two main subjects in one paper. The title of the paper is “ECSS: High Embedding Capacity Audio Watermarking with Diversity Reception” and the content should be only about this topic.

The definition of new terms “audio and signal” for various representations of the information is not needed and is confusing. The authors should use the consecrated terms.

The authors state “As copyright content becomes diverse, copyright owners want to embed the licensed object, the class of license and the start and end date into the host data, leading to the embedding capacity or payload gradually becoming a bottleneck of watermarking system.” This statement is undocumented. There is not necessary a need for increased capacity of the payload. The watermark can be a unique identifier for all the necessary data to be inserted in the material. If these data can be uniquely found based on the short identifier, the data can be stored somewhere else. Therefore, it is not necessary to store many data in the watermarked material in the scenario of copyright protection. Therefore, in copyright protection, the robustness is the main desired characteristic, and not data capacity. Data capacity becomes important in steganography scenarios.

The authors state “By extending the codebook, each sequence can carry more than one watermark bit, which improves the embedding capacity with guaranteed watermark robustness.” It is unclear what guaranteed robustness means. If it means 100% recovery under any circumstances, it is a false statement.

The authors state “This mechanism repeatedly embeds the watermark into different regions of the host audio and aggregates HSI from all repeated regions when extracting the watermark, reducing the randomness of HSI and achieving higher embedding capacity and stronger robustness.” It is unclear how the principle of inserting the same information repeatedly in a material can increase the capacity. The capacity should mean the quantity of non-repeated data that can be inserted in the material. If the same information is inserted repeatedly in a material, the capacity decreases while the robustness may increase.

“In this paper, the embedding capacity of the VSS watermarking algorithm is analyzed using information theory, where a Gaussian distribution is employed to build a statistical model of the HSI.” This is another topic and should be published separately. Therefore, section 3 should be removed.

The main topic of the paper begins in section 6, therefore the paper is very difficult to read with this many auxiliar details. The watermark algorithm is very briefly described, and the description may be incomplete. There is no information about the framing detection. Also, “The watermarked signal vectors are substituted in the original DCT coefficients, and then the watermarked audio is obtained by performing an inverse DCT and reconstructing the audio frame.” The SWR is given as an objective imperceptibility metric. Limits of this metric should be highlighted and evaluation on sections (i.e., 20 ms frames) should be provided to extract the lowest values obtained.

The comparison with other works is done in only one particular case. The performance should be presented for the whole broad collection of parameters that were used previously when ECSS was characterized.

Most important, the authors should treat the case of a concentrated attack: the case in which the attacker knows the algorithm and modifies the signal only where the watermark is inserted, as shown in section 6. The SWR can be recalculated after the attack. The authors should secure the algorithm in a way that an attacker, knowing the algorithm, cannot modify/remove the watermark. Crop attacks and robustness to framing errors should be evaluated. The effect of simple audio effects, like very short delay, discreet reverb, and other unnoticeable distortions help highlight the real-world usability of the algorithm.
